# The Relationship between Dentofacial Vertical Pattern and Bite Force Distribution among Children in Late Mixed Dentition

**Deema Ali AlShammery [1], Ahmad Mahdi AlShuruf [2], Nasser AlQhtani [3] and Sharat Chandra Pani [4],***

1 Division of Orthodontics, Riyadh Elm University, Riyadh 11681, Saudi Arabia; deema@riyadh.edu.sa
2 Department of Preventive Dental Sciences, Riyadh Elm University, Riyadh 11681, Saudi Arabia; ahmad.m.alshuhuf@student.riyadh.edu.sa
3 Department of Oral and Maxillofacial Surgery and Diagnostic Sciences, College of Dentistry, Prince Sattam Bin Abdulaziz University, AlKharj 11942, Saudi Arabia; Dr.nasser@live.co.uk
4 Division of Paediatric Dentistry and Orthodontics, Schulich School of Medicine and Dentistry, University of Western Ontario, London, ON N6A3K7, Canada
* Correspondence: spani@uwo.ca

**Featured Application: Digital bite measurement systems can be used to monitor changes in growth and development by clinical orthodontists.**

**Abstract:** Background: Digital bite measurement systems such as the T-Scan III allow for the computerized measurement of occlusal force distribution. This study aimed to establish the relationship between dentofacial vertical pattern and bite force distribution among children in late mixed dentition. Materials and Methods: In total, 86 children (45 male, 41 female) aged between 9 and 11 years with short ($n = 28$), medium ($n = 28$), and long ($n = 30$) facial heights were included in this study. The height, weight, age, and gender were recorded. Occlusal bite force distribution and time of occlusal cycle were recorded using a T-Scan III device (Tekscan Corp. Boston, MA, USA). The bite force distribution was compared among facial types using a One-Way ANOVA and post hoc test, a linear regression model with time of occlusion as dependent variable was developed. Results: No significant differences were observed in occlusion time between genders. Children with long facial height had a significantly lower anterior bite force distribution ($p < 0.05$) and significantly higher posterior bite force distribution ($p < 0.05$) than those with average or short facial height. Age, gender, height, and weight had no significant association with time of the occlusal cycle. Conclusion: Children with an increased vertical facial height have a more posterior distribution of force than children with average or short facial heights in the late mixed dentition.

**Keywords:** occlusion; vertical facial height; T-scan III; digital occlusal analysis

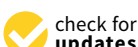



## 1. Introduction

When classified according to dentofacial vertical pattern there are three basic kinds of facial morphology: short, average, and long [1–3]. Many with a long face have excessive vertical facial growth that is generally associated with an open anterior bite, increased angle of sella-nasion (SN)/mandibular plane (MP), increased gonial angle, and increased angle of maxillary/mandibular planes [4]. Differences in jaw rotation between short-face and long-face subjects may also lead to changes in muscle force axes and different stimulation of muscle spindles of elevator muscles [5]. It has been postulated that if the bite force is considered a key determinant of masticatory function, then craniofacial morphology will also be expected to affect mastication [6].

While the mixed dentition period varies among children based on gender and ethnicity, it usually lasts between 9 and 11 years [7]. The stage also coincides with the pre-pubertal growth spurt, a time when the development of the dentofacial vertical pattern is established [8–10].

Bite force has long been of interest, as it is an indicator of the functional state of the masticatory system [11,12]. In building purpose and stability in the correction of malocclusions, an exact diagnosis remains the most venerable cornerstone [13]. There have been several studies that have attempted to measure the bite force of children in the mixed dentition and some have even attempted to study the impact of vertical dentofacial patterns on bite force [12,14–16]. However, recent research has shown that the pattern of distribution of bite forces is perhaps of far greater value than bite force alone [17,18].

The T-Scan system comprises a piezoelectric foil sensor, sensor handle and cable, system device, and data recording and analysis software [19]. The system was originally designed to help clinicians distinguish premature interactions, elevated powers, and occlusal surface interrelationships [20]. Recently there have been studies that have demonstrated the value of this system in the categorization and measurement of bite force patterns in children [17,21].

This study aimed to establish the relationship between dentofacial vertical pattern and bite force distribution among children in late mixed dentition.

## 2. Materials and Methods

### 2.1. Ethical Considerations

Ethical approval for the study was obtained from the Institutional Review Board of the Riyadh Elm University (FPGRP/2020/486/282/276). Informed consent was obtained from the parents of all participants and assent was taken from all participating children before examination.

### 2.2. Study Design

The study used a cross sectional study design.

### 2.3. Sample Power Calculation

Sample size was calculated using the G-Power sample power calculator (Universtat Kiel, Kiel, Germany). It was observed that for a medium effect size (0.5) and Beta of 0.95, with an alpha of 0.05, a sample of 85 children was required. A total of 86 children assented to the procedure thus giving a post hoc power of 0.956.

### 2.4. Inclusion and Exclusion Criteria

The sample comprised medically fit children aged between 9 and 11 years at the time of sampling. The children who participated in the study had erupted permanent incisors and molars, as well as primary molars and canines in both upper and lower arches. In order to avoid the confounding effect of molar relationship on bite pattern, only children with Class I occlusion were included in the study. Children with partially erupted permanent teeth or shed deciduous teeth were excluded from the study. Children with history of dental abscesses, restored teeth, or history of TMJ trauma were excluded from the study. Children with gross facial asymmetry or deviation of the jaw on mouth opening were also excluded from the study. All included participants were categorized as having a short, average, and long face based on previously established criteria based on the Mandibular Plane Angle (MPA) (1–3). A search of the cephaplometric records of age appropriate children from the databases of three different tertiary care centers was undertaken to screen a total of 90 children (30 per group) having a short ((MPA $< 28°$), normal ($28° \leq MPA \leq 39°$), and long (MPA $> 39°$) facial types.

### 2.5. Recording of Bite Force and Pattern and Mouth Opening

The bite force, and pattern were recorded using the T-Scan system and a small size sensor (Tekscan, Inc., Boston, MA, USA). The sensor was calibrated for each child by taking three measurements of the bite at centric relation. Each child was asked to bite on a sensor firmly and repeat this movement five times. Bite was recorded on a multi-bite analysis and the software calculated the contact position when the child came into

the maximum intercuspation position (MIP) (Figure 1). This process was repeated three times. The bite was then calibrated by using the protocol for children developed by Gallagher et al. [21]. The mean percentage of force on the first permanent molar was used to check the reproducibility of the bite.

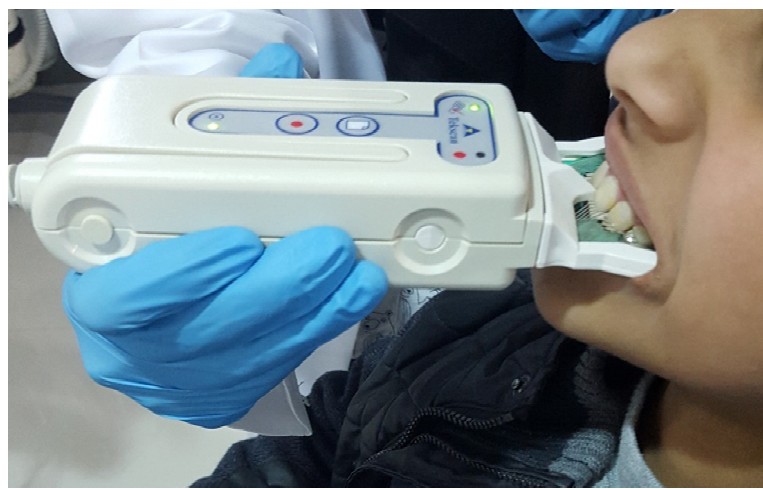

**Figure 1.** Measurement of the bite in a child using the T-Scan III system.

All measurements were performed by a single examiner (AS). A random re-examination and re-measurement of eight participants (10%) of the studied sample were carried out for occlusal force. Intra-examiner reproducibility was tested using Kappa statistic with a high degree of agreement (0.854).

*2.6. Statistical Analyses*

A Shapiro–Wilk test was performed to determine the normality of continuous variables such as Height ($p = 0.231$), Weight ($p = 0.871$), Occlusion Time ($p = 0.431$), and Bite Force ($p = 0.091$) and showed these variables to be normally distributed. These variables were compared among the groups using the One-Way ANOVA with a Scheffe's post hoc test. A linear regression model using occlusion time as a dependent variable and gender, height, weight, and facial type was developed. All analyses were carried out using the SPSS ver.25 data processing software (IBM-SPSS, Chicago, IL, USA).

**3. Results**

The sample comprised a total of 86 children (45 male, 41 female) aged between 9 and 11 years. The sample was selected so as to represent those with short ($n = 28$), medium ($n = 28$), and long ($n = 30$) facial heights. There were no significant differences in the age, weight, or height of the sample (Table 1). When the time taken for an occlusal cycle was compared among facial height types it was observed that although children with short and long facial height had a longer occlusal cycle, the differences were not statistically significant ($p = 0.322$).

When gender differences in the height and weight were compared, it was observed that there were no significant differences between males and females in either height or weight (Table 2). The same was observed in terms of percentage of occlusal force distribution in either of the four quadrants. It was therefore decided to compare the differences in force distribution among facial height types together for both genders.

**Table 1.** Distribution and descriptive data summary of the participants according to facial type.

| | Facial Height | | | |
|---|---|---|---|---|
| | **Short** | **Average** | **Long** | **Comparison** |
| Sample | $n = 28$ [15 Male, 13 Female] | $n = 28$ [14 Male, 14 Female] | $n = 30$ [16 Male, 14 Female] | |
| Age (years) | Range: 9–11 Mean = 9.2 SD = 1.5 | Range: 9–11 Mean = 8.9 SD = 1.6 | Range: 9–11 Mean = 9.1 SD = 1.3 | CI = −1.7015 to −0.9985 $p = 0.284$ |
| Weight (kg) | Range: 24–51 Mean = 33 SD = 10.4 | Range: 23–52 Mean = 29.5 SD = 7.9 | Range: 25–49 Mean = 28.5 SD = 6.7 | CI = −8.2730 to 1.2730 $p = 0.1476$ |
| Height (cm) | Range: 118–146 Mean = 129.1 SD = 7.1 | Range: 105–145 Mean = 128.8 SD = 6.8 | Range: 115–142 Mean = 127.6 SD = 6.5 | CI = −3.8145 to 3.2145 $p = 0.8649$ |
| Occlusion Time (second) | Range: 0.11–2.6 Mean = 0.75 SD = 0.72 | Range: 0.15–1.9 Mean = 0.68 SD = 0.81 | Range: 0.15–2.3 Mean = 0.76 SD = 0.69 | $p = 0.3211$ |

**Table 2.** Comparison of physical characteristics and bite characteristics between males and females.

| Group Statistics | Gender | Mean | Std. Deviation | t * | *p*-Value |
|---|---|---|---|---|---|
| Body height | Male Female | 138.2 136.7 | 6.7 5.8 | 1.086 | 0.281 |
| Body weight | Male Female | 32.8 33.7 | 4.9 6.0 | 0.807 | 0.422 |
| % distribution of occlusion right side anterior | Male Female | 13.2 13.8 | 6.3 6.6 | 0.402 | 0.688 |
| % distribution of occlusion right side posterior | Male Female | 39.1 38.0 | 9.3 9.2 | 0.544 | 0.588 |
| % distribution of occlusion left side anterior | Male Female | 11.9 12.5 | 6.9 7.5 | 0.361 | 0.719 |
| % distribution of occlusion left side posterior | Male Female | 35.8 35.7 | 5.0 5.9 | 0.072 | 0.943 |

* Calculated using the independent samples t test. There were no significant differences between males and females.

The analysis system recorded biteforce measurements at four different positions for all children (Table 3). The mean distribution of bite force recorded in the anterior right position for the short face with deep bite was 19.09 (SD = 1.47, Min = 16.0 and Max = 22.1). However, for the right posterior mean was 31.38 (SD = 1.85, Min = 27.20, and Max = 36.10) (Table 2). The One-Way ANOVA showed significant differences in bite force distribution among facial types in each of the four positions. In the anterior the children with a long facial height had a significantly lower bite force distribution than those with average or short facial height on both the left ($p = 0.0048$) and right ($p = 0.0003$) sides. The Scheffe's post hoc test showed that this difference was not significant between children who had a short face and those with average faces on either the left ($p = 0.124$) or right ($p = 0.342$) side.

When the bite forces in the posterior region were compared, it was observed that children with a long facial height had significantly greater posterior bite force distribution than children with average or short facial height on both the right ($p = 0.0006$) and left ($p = 0.0053$).

A linear regression model with the occlusal time as dependent variable found that neither age, weight, height, or facial type had any significant association with the occlusal time (Table 4).

**Table 3.** Occlusal bite force assessment comparison among facial types in the four different positions.

| Occlusal Force Distribution (%) | Facial Height | Mean (SD) | Sig. * |
|---|---|---|---|
| Anterior Right | Short [a] | 19.09 (1.47) | 0.0048 ** |
| | Average [a] | 16.96 (1.42) | |
| | Long [b] | 5.2 (1.51) | |
| Anterior Left | Short [a] | 17.68 (2.44) | 0.0003 ** |
| | Average [a] | 16.78 (1.73) | |
| | Long [b] | 2.99 (1.09) | |
| Posterior Right | Short [a] | 25.4 (1.7) | 0.0006 ** |
| | Average [a] | 35.1 (1.9) | |
| | Long [b] | 43.2 (7.9) | |
| Posterior Left | Short [a] | 31.86 (1.74) | 0.0053 ** |
| | Average [a] | 33.1 (2.74) | |
| | Long [b] | 41.71 (8.8) | |

* Calculated using One-Way ANOVA. ** Differences significant at $p < 0.05$. a, b: differences in superscript indicate significant differences at $p < 0.05$ when measured using the Scheffe's post hoc test.

**Table 4.** Regression model of factors influencing the occlusal time.

| Group | Model | Unstandardized Coefficients | | Standardized Coefficients | t | Sig. |
|---|---|---|---|---|---|---|
| | | B | Std. Error | Beta | | |
| | (Constant) | 0.174 | 0.539 | | 0.323 | 0.750 |
| | Weight | 0.000 | 0.009 | 0.029 | 0.019 | 0.985 |
| | Height | −0.001 | 0.004 | −0.172 | −0.283 | 0.780 |
| | Facial Type | −123 | 0.243 | −132 | −4.31 | 0.231 |
| | Age | −0.028 | 0.116 | −0.055 | −0.239 | 0.813 |

No significant associations with occlusal time.

## 4. Discussion

Although the relationship of bite force to age and gender and facial type has been previously studied, the exact impact of facial height on bite force distribution has remained relatively unexplored [22]. This study sought to compare the distribution of occlusal forces among children with short, average, and long facial heights.

The development of digital bite analysis software has allowed researchers to measure not only bite force, but also map the distribution of the bite and the impact this has on the time of the occlusal cycle [23]. The fact that bite pattern is a better indicator of function than only bite force is a concept that has been gaining traction in literature [17,18]. The fact that there was no significant difference between males and females in the distribution of occlusal forces is in keeping with studies that show that despite gender differences in force, there is no difference in the pattern of distribution of forces. Similarly, the fact that there was no influence of age, body weight, or body height to the distribution of the bite is in keeping with studies that show that bite pattern may be independent of bite force [23,24].

Studies have previously sought to measure the impact of facial height on bite force and maximum mouth opening [25,26]. However, the impact of facial type on the distribution of the bite has remained relatively unexplored. The findings of the current study found that while there was no difference between children with a short facial height and average facial height, children with a long facial height had a significantly greater distribution of force in the posterior segments and significantly lower distribution of bite in the anterior segment. Despite this finding, there were no significant differences among the three facial types in the time of the occlusal cycle. This is important, as increases in the time of the occlusal cycle are suggestive of loss of balanced occlusion [18]. This would seem to suggest that although children in the late mixed dentition with an increased facial height have a

greater posterior distribution of force, there is no significant imbalance in the occlusion when compared to those with average or short facial heights.

The results of this study should be viewed keeping in mind certain limitations. The study only focused on children with Class I occlusion. Furthermore, the study did not take into consideration anomalies in bite and focused only on the distribution of force. The fact that this study was done on patients who were awaiting orthodontic treatment and the fact that a habitual chewing side was not considered in the selection of patients is a further limitation of the study. Future studies need to explore the more complicated relationship between the sagittal, transverse, and vertical occlusal relationships.

## 5. Conclusions

Within the limitation of the current study, we can conclude in the late mixed dentition, children with an increased vertical facial height have a more posterior distribution of force than children with average or short facial heights.

**Author Contributions:** Conceptualization, D.A.A. and N.A.; methodology, S.C.P. and D.A.A.; software, A.M.A. and D.A.A.; validation, A.M.A., N.A. and S.C.P.; formal analysis, S.C.P.; investigation, A.M.A.; resources, D.A.A. data curation, D.A.A. and A.M.A.; writing—original draft preparation, A.M.A. and N.A.; writing—review and editing, S.C.P. All authors have read and agreed to the published version of the manuscript.

**Funding:** This study was self funded.

**Institutional Review Board Statement:** The study was conducted according to the guidelines of the Declaration of Helsinki and approved by the Institutional Review Board of Riyadh Elm University (FPGRP/2020/486/282/276).

**Informed Consent Statement:** Informed consent was obtained from the parents of all participants and assent was taken from all participating children before examination.

**Data Availability Statement:** Supplementary data files will be made available upon reasonable request to the authors.

**Acknowledgments:** The authors would like to acknowledge the Ministry of Health, Kingdom of Saudi Arabia, for sponsoring the postgraduate training of AlShuruf.

**Conflicts of Interest:** The authors declare no conflict of interest.

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
