# Peer review of "The Relationship between Dentofacial Vertical Pattern and Bite Force Distribution among Children in Late Mixed Dentition"

_applsci, doi:10.3390/app112110140_

Round 1
Reviewer 1 Report
The paper lacks academic value. I suggest rejecting the manuscript.
Author Response
We thank the reviewer for thier time. We were unable to find specific comments to respond to
Reviewer 2 Report
i suggest to authors :
-to provide reference for the afirmation : "All included participants were categorized as having a short, average, and long face based on previously established criteria"-page 2 raw 81
-to present the limits of digital bite analysis software and its proved validity
-to complete the pages for the reference 8,14,18,21
-
Author Response
Major concerns: 1.The authors claimed that the age, between 9 and 11 years, coincided with the pre-pubertal growth spurt when the development of dentofacial vertical pattern was established, but different genders may affect the time of growth spurt. Besides, growth in vertical height of face could last very long. Whether these factors will reduce soundness should be reconsidered.
We agree with the reviewer. The sample was however random in their selection and the fact that there was no significant difference in demographic factors among the groups would control for such variations. We have addressed this concern in the discussion
2.In inclusion and exclusion criteria, the authors categorized participants as having short, average and long faces based on “previously established criteria”. The specific classification of different dentofacial vertical patterns should be described.
A description of the criteria has been added to the materials and methods section
3.Though the authors concluded that there was no significant imbalance in occlusion between those with long facial heights and average or short facial heights, the study only included children with Class I occlusion. Whether dentofacial types will affect the results in different occlusion was not considered.
We agree with the reviewer, there is evidence in literature that different occlusal types affect bite pattern. In order to rule out the confounding effect of occlusal type, this study was restricted to individuals in Class I occlusion. We agree that the investigation of other types of occlusion are a matter for further study and this has been included in the investigation.
Minor concerns:
1.The arrangement of table 4 should be corrected.
The table arrangement has been corrected
2.The format and punctuation should be double checked.
The format and punctuation have been double checked
Point 1 [15–29]: In the abstract paragraph numbers should be removed. The number of
spaces should be corrected - in the text there are double spaces and their complete absence.
"T-scan III" should be recorded with a space between "T-scan" and "III" [14, 21,
30 ...].
We thank the reviewer the changes have been made
Point 2 [25, 26]: The statistical significance of the study was set at 5% that is why p must
be indicated for 0.05, not 0.01. Moreover, in the results (Table 3) p <0.05 is only for the
Anterior Left and Posterior Right localizations; p value for Anterior Right, Posterior Left
locations is ambiguous (close to 0.05). Despite this, in the results described in the abstract
you presented generalized data which is incorrect.
The data has been re-described in the table and reformatted as per the suggestion of reviewer 2. Please note the significance is 0.0048 and 0.0053 (which is less than 0.01)NOT 0.048 or 0.053 as the reviewer seems to imply!
Point 3: Inline bibliographic references do not meet the requirements of the MDPI Reference
List and Citations Style Guide (Link). Please pay attention to the spaces between
the text and links.
Noted this has been edited throughout the manuscript
Point 4 [63–66]: The information about Study design isn’t provided.
This has been added
Point 5 [70]: The calculated required sample size is 85. Why were 86 people included in
the study?
The minimum required sample size was 85. Given the age group and the chance that a reliable bite may not be possible a total of 90 (30 per group) were recruited. The total acceptable bites resulted in a sample of 86. This has been added to section on calibration regarding the t-scan and to the methodology
Point 6 [86]: Missing point in line.
Point 7 [84]: The T-scan measurement technique should be described more detailed to
ensure that your study is representative.
- How did you calibrate the sensor's sensitivity?
- Has the sensor surface morphology been pre-adapted?
- What type of closure was used for measurements (multi-bite, single-bite, extrusion)?
- What time point in the recording of the closure of the dentition did you take for the
final evaluation of the results?
- What sensor size did you use in your study?
- “S” size sensors are suitable for most adults. I'm not sure that a sensor of this size can
be positioned in the oral cavity of pediatric patients. Please provide a photograph of the
sensor positioned in your patient's mouth using the T-scan.
The sensor was calibrated for each child by taking three measurements of the bite at centric relation. Each child was asked to bite on a sensor firmly and repeat this movement 5 times. Bites was recorded on a multi-bite analysis and the software calculated the contact position when the child came into the maximum intercuspation position (MIP) (Figure 3). This process was repeated three times. The bite was then calibrated by using the protocol developed by Gallagher et al.(2014). The mean percentage of force on the first permanent molar was used to check the reproducibility of the bite. Variation of greater than 10% in bite force were considered to be unacceptable and the records of the patient were excluded from the study.
Point 8 [90]: As ANOVA is a parametric method It is necessary to describe the procedure
of the data normal distribution checking.
We agree the Shapiro Wilk test values have been added
Point 9 [120 (Table 3)]: Values for Long - Posterior Right are missing in the table.
This table is formatted incorrectly and data is missing. Based on the recommendation from reviewer 2 the table has been formatted.
Point 10 [120 (Table 3)]: According to the inclusion and exclusion criteria presented in
the article, the formed groups should not include children with severe pathologies of the
dentition. However, Table 3 shows the data that indicate de-occlusion for one of the patients
with short facial height in the Posterior Right side . This is a contradiction. Considering
this fact, as well as the revealed significant differences only in two groups out of
four, the inputs indicated in the article seem doubtful.
The issue regarding table 3 has been clarified. Please note your assumptions of significance among two groups are incorrect – note that p value has been calculated up to four decimal places not three. This p for the groups you mention is 0.0048 not 0.048 and 0.0053 not 0.053
Point 11 [141 (Table 4)]: The data is carelessly indicated in the table.
We apologize the table has been formatted
Group |
Model |
Unstandardized Coefficients |
Standardized Coefficients |
t |
Sig. |
|
B |
Std. Error |
Beta |
||||
|
(Constant) |
.174 |
.539 |
|
.323 |
.750 |
Weight |
.000 |
.009 |
.029 |
.019 |
.985 |
|
Height |
-.001 |
.004 |
-.172 |
-.283 |
.780 |
|
Facial Type |
-123 |
.243 |
-132 |
-431 |
.231 |
|
Age |
-.028 |
.116 |
-.055 |
-.239 |
.813 |
Point 12: In this article the issue of habitual chewing side that could argue and clarify the
findings of the study isn’t covered. Was this taken into account in the selection of the
samples? If yes then it is worth to clarify in the “Materials and Methods” (Inclusion /
Exclusion Criteria) and “Discussion” sections. If no then it is worth to designate in "research
limitations".
A paragraph on this has been added to the limitations of the study
Reviewer 3 Report
The manuscript titled “The Relationship between Dentofacial Vertical Pattern and Bite Force Distribution among Children in Late Mixed Dentition” evaluated the relationship between dentofacial vertical pattern and bite force distribution among children in late mixed dentition. In this study, the authors determined that children with an increased vertical facial height had a more posterior distribution of force than children with average or short facial heights in late mixed dentition. This is an interesting observation to study the relationship between dentofacial type and occlusal patterns. However, additional explanations are required to further illustrate the different bite force distribution of children with long, average and short facial heights. Major concerns: 1.The authors claimed that the age, between 9 and 11 years, coincided with the pre-pubertal growth spurt when the development of dentofacial vertical pattern was established, but different genders may affect the time of growth spurt. Besides, growth in vertical height of face could last very long. Whether these factors will reduce soundness should be reconsidered. 2.In inclusion and exclusion criteria, the authors categorized participants as having short, average and long faces based on “previously established criteria”. The specific classification of different dentofacial vertical patterns should be described. 3.Though the authors concluded that there was no significant imbalance in occlusion between those with long facial heights and average or short facial heights, the study only included children with Class I occlusion. Whether dentofacial types will affect the results in different occlusion was not considered. Minor concerns: 1.The arrangement of table 4 should be corrected. 2.The format and punctuation should be double checked.Author Response
Major concerns: 1.The authors claimed that the age, between 9 and 11 years, coincided with the pre-pubertal growth spurt when the development of dentofacial vertical pattern was established, but different genders may affect the time of growth spurt. Besides, growth in vertical height of face could last very long. Whether these factors will reduce soundness should be reconsidered.
We thank the reviewer for this observation. We agree that there could be issues with the timing of the growth spurt, we calculated the
2.In inclusion and exclusion criteria, the authors categorized participants as having short, average and long faces based on “previously established criteria”. The specific classification of different dentofacial vertical patterns should be described.
A description of the criteria has been added to the materials and methods section
3.Though the authors concluded that there was no significant imbalance in occlusion between those with long facial heights and average or short facial heights, the study only included children with Class I occlusion. Whether dentofacial types will affect the results in different occlusion was not considered.
We agree with the reviewer, there is evidence in literature that different occlusal types affect bite pattern. In order to rule out the confounding effect of occlusal type, this study was restricted to individuals in Class I occlusion. We agree that the investigation of other types of occlusion are a matter for further study and this has been included in the investigation.
Minor concerns:
1.The arrangement of table 4 should be corrected.
The table arrangement has been corrected
2.The format and punctuation should be double checked.
The format and punctuation have been double checked
Reviewer 4 Report
Dear Authors!
Comments in the attached file.
The article requires serious improvements.

Author Response
Point 1 [15–29]: In the abstract paragraph numbers should be removed. The number of
spaces should be corrected - in the text there are double spaces and their complete absence.
"T-scan III" should be recorded with a space between "T-scan" and "III" [14, 21,
30 ...].
We thank the reviewer the changes have been made
Point 2 [25, 26]: The statistical significance of the study was set at 5% that is why p must
be indicated for 0.05, not 0.01. Moreover, in the results (Table 3) p <0.05 is only for the
Anterior Left and Posterior Right localizations; p value for Anterior Right, Posterior Left
locations is ambiguous (close to 0.05). Despite this, in the results described in the abstract
you presented generalized data which is incorrect.
The data has been re-described in the table and reformatted as per the suggestion of reviewer 2. Please note the significance is 0.0048 and 0.0053 (which is less than 0.01)NOT 0.048 or 0.053 as the reviewer seems to imply!
Point 3: Inline bibliographic references do not meet the requirements of the MDPI Reference
List and Citations Style Guide (Link). Please pay attention to the spaces between
the text and links.
Noted this has been edited throughout the manuscript
Point 4 [63–66]: The information about Study design isn’t provided.
This has been added
Point 5 [70]: The calculated required sample size is 85. Why were 86 people included in
the study?
The minimum required sample size was 85. Given the age group and the chance that a reliable bite may not be possible a total of 90 (30 per group) were recruited. The total acceptable bites resulted in a sample of 86. This has been added to section on calibration regarding the t-scan and to the methodology
Point 6 [86]: Missing point in line.
Point 7 [84]: The T-scan measurement technique should be described more detailed to
ensure that your study is representative.
- How did you calibrate the sensor's sensitivity?
- Has the sensor surface morphology been pre-adapted?
- What type of closure was used for measurements (multi-bite, single-bite, extrusion)?
- What time point in the recording of the closure of the dentition did you take for the
final evaluation of the results?
- What sensor size did you use in your study?
- “S” size sensors are suitable for most adults. I'm not sure that a sensor of this size can
be positioned in the oral cavity of pediatric patients. Please provide a photograph of the
sensor positioned in your patient's mouth using the T-scan.
The sensor was calibrated for each child by taking three measurements of the bite at centric relation. Each child was asked to bite on a sensor firmly and repeat this movement 5 times. Bites was recorded on a multi-bite analysis and the software calculated the contact position when the child came into the maximum intercuspation position (MIP) (Figure 3). This process was repeated three times. The bite was then calibrated by using the protocol developed by Gallagher et al.(2014). The mean percentage of force on the first permanent molar was used to check the reproducibility of the bite. Variation of greater than 10% in bite force were considered to be unacceptable and the records of the patient were excluded from the study.
Point 8 [90]: As ANOVA is a parametric method It is necessary to describe the procedure
of the data normal distribution checking.
We agree the Shapiro Wilk test values have been added
Point 9 [120 (Table 3)]: Values for Long - Posterior Right are missing in the table.
This table is formatted incorrectly and data is missing. Based on the recommendation from reviewer 2 the table has been formatted.
Point 10 [120 (Table 3)]: According to the inclusion and exclusion criteria presented in
the article, the formed groups should not include children with severe pathologies of the
dentition. However, Table 3 shows the data that indicate de-occlusion for one of the patients
with short facial height in the Posterior Right side . This is a contradiction. Considering
this fact, as well as the revealed significant differences only in two groups out of
four, the inputs indicated in the article seem doubtful.
The issue regarding table 3 has been clarified. Please note your assumptions of significance among two groups are incorrect – note that p value has been calculated up to four decimal places not three. This p for the groups you mention is 0.0048 not 0.048 and 0.0053 not 0.053
Point 11 [141 (Table 4)]: The data is carelessly indicated in the table.
We apologize the table has been formatted
Group |
Model |
Unstandardized Coefficients |
Standardized Coefficients |
t |
Sig. |
|
B |
Std. Error |
Beta |
||||
|
(Constant) |
.174 |
.539 |
|
.323 |
.750 |
Weight |
.000 |
.009 |
.029 |
.019 |
.985 |
|
Height |
-.001 |
.004 |
-.172 |
-.283 |
.780 |
|
Facial Type |
-123 |
.243 |
-132 |
-431 |
.231 |
|
Age |
-.028 |
.116 |
-.055 |
-.239 |
.813 |
Point 12: In this article the issue of habitual chewing side that could argue and clarify the
findings of the study isn’t covered. Was this taken into account in the selection of the
samples? If yes then it is worth to clarify in the “Materials and Methods” (Inclusion /
Exclusion Criteria) and “Discussion” sections. If no then it is worth to designate in "research
limitations".
A paragraph on this has been added to the limitations of the study
Round 2
Reviewer 1 Report
The revision of the paper does not solve the previous problems
Reviewer 3 Report
The authors addressed all of my questions and concern
Reviewer 4 Report
Inline bibliographic references should be in square brackets with space between text and brackets.